# Do Out-of-Pocket Payments for Care for Children under 5 Persist Even in a Context of Free Healthcare in Burkina Faso? Evidence from a Cross-Sectional Population-Based Survey

**DOI:** 10.3390/healthcare11101379

**Published:** 2023-05-10

**Authors:** Ludovic D. G. Tapsoba, Mimbouré Yara, Meike I. Nakovics, Serge M. A. Somda, Julia Lohmann, Paul J. Robyn, Saidou Hamadou, Hervé Hien, Manuela De Allegri

**Affiliations:** 1Centre MURAZ, National Institute of Public Health, Bobo-Dioulasso 390, Burkina Faso; 2Heidelberg Institute of Global Health, Medical Faculty and University Hospital, University of Heidelberg, 69120 Heidelberg, Germany; 3UFR Exact and Applied Sciences, Nazi Boni University, Bobo-Dioulasso BP 1091, Burkina Faso; 4Department of Global Health and Development, London School of Hygiene & Tropical Medicine, Keppel Street, London WC1E 7HT, UK; 5The World Bank Group, 1818 H St. NW, Washington, DC 20433, USA

**Keywords:** free healthcare, out-of-pocket payments, children under 5, Burkina Faso

## Abstract

Background: In April 2016, Burkina Faso began free healthcare for children aged from 0 to 5 years. However, its implementation faces challenges, and the goal of this study is to estimate the fees paid for this child care and to determine the causes of these direct payments. Methods: Data gathering involved 807 children aged from 0 to 5 years who had contact with the public healthcare system. The estimation of the determinants of out-of-pocket health payments involved the application of a two-part regression model. Results: About 31% of the children made out-of-pocket payments for healthcare (an average of 3407.77 CFA francs per case of illness). Of these, 96% paid for medicines and 24% paid for consultations. The first model showed that out-of-pocket payments were positively associated with hospitalization, urban area of residence, and severity of illness, were made in the East-Central and North-Central regions, and were negatively associated with the 7 to 23 month age range. The second model showed that hospitalization and severity of illness increased the amount of direct health payments. Conclusion: Children targeted by free healthcare still make out-of-pocket payments. The dysfunction of this policy needs to be studied to ensure adequate financial protection for children in Burkina Faso.

## 1. Background

An increasing number of countries in Sub-Saharan Africa have recently implemented policies to waive/reduce the cost of healthcare for the poorest members of their populations or for specific sections of the populations such as pregnant women and children under five [1]. These policies vary from country to country in terms of the services covered and the social groups benefiting. For example, in 2006, Senegal introduced a policy of eliminating user fees for childbirth care at the national level; in 2005 Mali introduced a policy of eliminating fees for cesarean sections. Several studies have shown that out-of-pocket payments can be a barrier to the use of health services [2,3]. Fee removal/reduction policies aim to increase access to healthcare facilities, thereby contributing to the reduction of maternal and neonatal mortality and helping to achieve the Sustainable Development Goals [4]. The adoption of these policies is not an easy task, as public policy development and implementation in most African countries are fraught with difficulties [5].

The studies of fee removal/reduction policies have mostly focused on the impact on healthcare service utilization rather than on financial protection. They show that these policies increase the use of healthcare services [6]. However, studies have shown that out-of-pocket health payments still remain in free care settings, but the studies are not sufficiently representative of the country’s population [7,8,9,10,11,12].

In Burkina-Faso, a series of measures to waive user fees have been taken since the 2000s. In 2005, the management of severe malaria cases was fully subsidized [13]. Impregnated mosquito nets have also been subsidized and distributed throughout the country since 2010. In the area of maternal health, a subsidy of 60 to 80%, depending on the services offered, is provided for complicated and elective deliveries. The Government of Burkina Faso has initiated the total exemption of healthcare fees for women and children aged from 0 to 5 years as of 1 June 2016 after successful pilot experiences. This free care strategy aims to significantly reduce preventable deaths among children aged from 0 to 5 years and women [14]. As part of this policy, many sub-Saharan countries are reimbursing health facilities [15]. However, Burkina Faso has started the financial management of this policy by pre-positioning funds for the covered services in accounts opened specifically for the free healthcare policy. This strategy was adopted to avoid delays in reimbursement, which encouraged out-of-pocket payments. Few studies have examined the drivers of out-of-pocket expenditure (OOPE) in a context of free care delivery [16,17].

This study aims at filling an existing gap in the literature as one of the first to examine the effects of the national free healthcare policy on out-of-pocket spending for child healthcare services. Prior impact evaluations have either referred to pilot experiences [18] or examined the impact on payments for maternal care services [16]. This study builds on household data from 24 districts to measure out-of-pocket expenditure for children aged 0 to 5 years more than one year after the launch of the free healthcare policy.

## 2. Methodology

### 2.1. Framework of the Study

Burkina Faso is a low-income country with an agricultural vocation. Its economy is subject to climatic hazards, fluctuations in world trade conditions, and the exchange rate (INSD, Demographic and Health Survey and Multiple Indicators “EDSBF-MICS IV”, 2010). The 2014 Continuous Multisectoral Survey (CMS) reveals that 40.1% of the Burkinabe population is poor [19]. As one of the poorest countries in the world, the country has been making economic progress in recent years thanks to relatively high annual economic growth (+6.51% in 2018). However, Burkina Faso has a long way to go on the road to development. The GDP per capita is only XOF444,817.96 (US$740.75 (1 XOF = 0.0017 US$ (April 2023)) (2018). This makes it difficult for people to access basic social services and increases unemployment. In addition, the country suffers from high morbidity and mortality, mainly due to acute infectious diseases (acute malaria, acute respiratory infections, diarrhea, etc.) and high infant and maternal mortality rates (World Health Organization, 2015; African Health Observatory, 2016). As a result, the country ranks 185 out of 188 countries on the 2016 Human Development Index (HDI) (United Nations Development Program, 2016) [20].

The healthcare structure is organized in three levels. The first level is the health district, which encompasses two parts. The first level includes the Health and Social Promotion Centers (HSPC), which is the first contact point for a wide range of primary care services for children aged 0–5 years. The second level is the Medical and Surgical Centers (MSC). In 2020, there were 70 health districts with 2041 Health and Social Promotion Centers (HSPC) and 46 Medical and Surgical Centers (MSC) in operation. In 2020, the second level included nine Regional Hospital Centers (RHC) serving as references for the CMAs, and the third level consists of six University Hospital Centers. In addition, Burkina Faso has 641 private facilities concentrated in the cities of Ouagadougou and Bobo-Dioulasso [21].

Out-of-pocket health payments accounted for 35.8% of current health expenditures in 2018 (National Health Accounts 2018). This figure is high by WHO standards (the WHO stipulates that the percentage of out-of-pocket payments on current health expenditures should not exceed 20%).

### 2.2. The Free Care Policy for Children from 0 to 5 Years

On 2 March 2016, Burkina-Faso adopted a policy of free care for children aged from 0 to 5 years, which was implemented on 2 April 2016, by the health districts of the Center, Sahel, and Hauts-Bassins regions. On 1 May 2016, the hospitals in the aforementioned regions also started the implementation. As of the 1 June 2016, the implementation was extended to all the other facilities in the other regions. Free care is offered in public health facilities and private health facilities that have signed an agreement with the Ministry of Health.

The targeted populations benefit from free care whatever the medical or surgical specialty.

Healthcare for children aged from 0 to 5 years is preventive, diagnostic, and curative in both outpatient and inpatient/observation settings for all common conditions targeted by the IMCI (Integrated Management of Childhood Illness) strategy to reduce infant and child mortality [22].

### 2.3. Data and Data Sources

Data for this study were gathered from the Final Household Survey for the impact evaluation of the Results-Based Financing (Fbr), financed by the World Bank between April and June 2017. The sample is composed of 7935 households in 24 districts (intervention and control) in the six project regions (Boucle of Mouhoun, North-Central, West-Central, North, South-West, and East-Central). These households were selected in two phases: in the first phase, villages were selected randomly from the health areas. One village was randomly selected for each HSPC. In the second phase, 15 households were randomly selected in each village based on a complete list of households. This list consisted of households with at least one pregnant woman or one woman who had given birth in the last 24 months.

For this study, we used the household module, namely the sections on socioeconomic and demographic characteristics of households, education, and health, particularly healthcare expenditures for acute illnesses. In each household, the respondents were the head of the household, pregnant women, or women who had given birth in the last 24 months, and all children aged from 0 to 5 years. Our study therefore focuses on children aged from 0 to 5 years.

### 2.4. Description of the Sample

We focused on children aged from 0 to 5 years who reported having a non-chronic illness or injury and whose parents sought care. The data allowed us to identify those children who reported contact with the formal public healthcare system for care as in previous studies [16]. This target was taken because we are interested in the free healthcare policy that is implemented in our study in public healthcare facilities in Burkina Faso. The identification steps of our sub-sample are visible in the flow chart (Figure 1).

### 2.5. Description of the Variables and Their Measures

#### 2.5.1. Variable of Interest

The “Out-of-pocket expenditure” is the main variable of interest. It aggregates all health expenditures made by the household for children aged from 0 to 5 who were sick in the last 24 months and who used healthcare in a public health facility. This package of care concerns only those in the free basket (including hospitalization) when a child aged 0–5 is cared for in a public health facility. These expenditures include consultation fees, laboratory/radiography and surgical fees (examinations), purchase of medicines, and hospitalization fees. Informal payments without receipts and transportation costs were therefore not included in the calculations of out-of-pocket health payments. Calculations were made at the individual, not the household, level. This avoids the risk of double-counting costs. Some of these costs might have been paid outside the health facility, namely those related to the purchase of drugs and examination fees. Expenditures were calculated in the local currency of the country, i.e., in XOF. Indirect costs were not considered.

#### 2.5.2. Explanatory (Control) Variables

The research of factors associated with out-of-pocket expenditure highlights the socio-demographic data of patients and their mothers (age, place of residence, mother’s level of education, economic status) and those related to the health system (health district, region, distance from the health center) as control variables (Table 1).

The “economic status” ranks households in the poorest quintile of socio-economic status (SES) against the other four richer quintiles. This approach follows the logic of the most vulnerable exemption policy that started its experimentation since 2008 with the collaboration of a German NGO (HELP). Socio-economic status (SES) was calculated as a continuous variable using the Multiple Component Analysis (MCA) method to aggregate information on assets, household housing characteristics, and other parameters, and was divided into quintiles [23,24].

We categorized “disease severity” into no effect on activities and some effect on activities (leisure, school, being able to eat, etc.).

### 2.6. Analytical Approach

Our objective was to identify how many children face an out-of-pocket expenditure, quantify its amount, and determine the associated factors. First, we computed mean values, standard deviation (SD), median, min and max values. To correct for outliers, we applied winsorizing, replacing 5 to 25% of outlier expenditure data items [25]. This method preserves the power of the sample in the regression analysis and avoids bias in the calculation of the mean value and variance by retaining higher values in the sample [26].

Second, we examined which factors were associated with a positive expenditure and with the magnitude of the expenditure. To address usual challenges related to the large number of zeros, skewness, and possible bias due to heteroscedasticity, several authors have recommended a two-part model approach [27]. Accordingly, the probability of facing a positive direct health payment was modeled first (Pr (Y > 0|X)) and followed by modeling the amounts of out-of-pocket health payments for those who paid (E (Y|Y > 0, X)) [27], as expressed in the equation below:E (Y|X) = Pr (Y > 0|X) × E (Y|Y > 0, X).

Hence, for the first part, a logit regression was done to identify the “probability of incurring OOPE” because the interpretation of its results is quite simple. For the second part, the Box-Cox test (*p* = 0 for λ = 0) showed that the data are not suitable for the use of the generalized linear model (GLM) with a log link. Therefore, the OLS model with log transformation of Y was used to identify the “determinants of out-of-pocket health payments” in the second part. For the gross scale back transformation, the restrictive assumption of normal distribution of log scale errors was not imposed. The estimator of Duan (1983) was used in preference [27,28]. The data analysis was performed with STATA version 13.

### 2.7. Ethical Considerations

The study received the favorable opinion of the ethics committee of the University of Heidelberg (S-272/2013) and the national ethics committee for health research of Burkina Faso (N° 2013-7-066 and N° 2015-5-071). Respondents gave their informed consent, and their confidentiality was ensured by the anonymity used in the data collection tools.

## 3. Results

### 3.1. Characteristics of Sample

The sample included a total of 15,323 children aged from 0 to 5 years (Figure 1). Of these, 1,015 children reported a non-chronic illness or injury in the four weeks preceding the survey date. Of these, 807 sick children aged 0 to 5 years had contact with the formal public healthcare system. Most of these children did not come from poor households and lived in rural areas (*n* = 807, 90.46%) with an average distance from the household to the health facility of less than or equal to 5 km. These children were from FBR districts for 79.18% and went to primary healthcare facilities (HSPC, maternity) to receive care (97.40%).

### 3.2. Direct Childcare Services Payments

About 31% (*n* = 248) of those who had contact with the formal public healthcare system faced direct childcare service payments that averaged XOF3407.77 (US$ 5.67) (min: 100 frs and max: 10,000 frs) per illness episode (Table 2). The highest average expenditure was observed for hospitalization (concerning 4%), laboratory/radiology and surgical services (concerning 2%), and consultations (concerning 24%). The most frequent expenditures were for medical drugs (96%) and for consultation fees (24%). Additional expenses outside the free care package were noted. These are transportation expenses for 109 children (44%) whose parents declared that they had paid for transportation out of their own pockets with an average of XOF728 (US$1.21). This gives an indication of the financial burden on households. The highest variance is for hospitalization (SD = XOF16,358.74) and for consultation (XOF3459.1)

### 3.3. Factors Associated with OOP Payments

The percentage of children aged from 0 to 5 years who paid for care differed significantly according to the child’s age group, severity of illness, economic status, place of residence, and region. The following factors have no impact on childcare service payments: mother’s level of education in the household, proximity to the health center, and district FBR. Table 3 presents the proportions of free and paid care services according to children aged from 0 to 5 years characteristics. This table allowed us to select the variables of the two-part model, taking variables as significant at the 20% level.

Table 4 presents the adjusted results of the two-part model that included 807 observations in the first part and 248 in the second part. In the logistic regression (first part), hospitalization, place of residence, region of residence, child’s age group, economic status, and severity of illness were significantly associated with out-of-pocket payments for children aged from 0 to 5 years. As a result, those who are hospitalized are 4.53 times more (95% CI= 1.92–8.88) likely to pay for childcare services than those not hospitalized. Compared with those in rural places, those in urban places paid for childcare services 3.47 times more often (95% CI = 1.88–5.34). People with an illness that affects any activity are 1.53 times more likely (95% CI = 1.07–2.09) to pay for childcare services than those whose illness does not affect their activities. Compared with the South-West region, those that lived in the East-Central and North-Central were 4.50 times (95% CI = 2.18–9.29) and 2.43 times (95% CI = 1.16–5.08) more likely to pay for childcare services for children aged from 0 to 5 years, respectively. In addition, children in the 7- to 23-month age group are 48 percent (95% CI= 0.35–0.83) less likely to pay for childcare services than those in the 0- to 6-month age group. The poorest section of the population (quintile 1) are 33 percent (95% CI= 0.44–1.07) less likely to pay for childcare services. Thus, the poor are protected.

In the log-transformed OLS regression (second part), hospitalization significantly increased the amount of out-of-pocket health payments for children aged from 0 to 5 years (coef = 0.78; *p* < 0.001). The amount was higher for people with an illness that affects any activity (coef = 0.24; *p* = 0.047). In addition, locations such as the East-Central, North-Central, and West-Central region saw increased the amounts of out-of-pocket health payments when there was a direct childcare services payment (coef= 0.60; 0.59; 0.76 and *p* = 0.029; 0.04; 0.01) compared with the South-West region.

## 4. Discussion

Our study is one of the first ones to estimate the level and determinants of out-of-pocket payments for care among children aged 0 to 5 years of age in the context of the gratuité, the free healthcare policy launched nationwide in 2016 in Burkina Faso. Our study makes an important contribution to the literature by providing evidence on the efficacy of the policy in removing user charges at point of use during its early implementation phase, building on a representative population-based sample and covering over a third of the country. Prior studies measuring OOP expenditure in the context of free healthcare policies in Burkina Faso have traditionally drawn from pilot experiences or focused on expenditures for maternal care [16,18].

First, our results show that one year into the implementation, one third of all children aged 0 to 5 years of age who reported an episode of illness or injury and who sought care in the formal public healthcare system incurred direct health payments in the process of seeking care. Among those who incurred a payment, the amount averaged XOF 3407.77 (US$5.67) per episode of care. This amount is considerable in a country where, even in the formal sector, the minimum wage is equivalent to XOF 34,664 and where 44% of the population lives in extreme poverty on less than US$2 per day [29]. While our study has not investigated how families cope with these high payments, we can only speculate that past strategies (including asset erosion and borrowing) persist and contribute to further household impoverishment [30,31]. Our results are consistent with what has been observed in prior studies, suggesting that the introduction of free healthcare policies alone is not sufficient to remove all payments at point of use [12,16,32,33]. Prior research has indicated that challenges related to the implementation of healthcare policies, including insufficient knowledge among implementers and funding, which are often responsible for residual fees [34,35,36]. Further research is needed to investigate the root causes of our observations.

Second, our analysis shows that hospitalization, laboratory/radiology services, and surgery, although infrequent events, were key cost drivers. This suggests a failure on behalf of the system to provide free healthcare consistently across all levels of care. Similar patterns have been observed elsewhere in the presence of free healthcare policies. For instance, in Zambia, almost all children accessed care free of charge after the introduction of the free healthcare policy, but nearly 10% were left to pay catastrophic amounts [9]. This suggests that healthcare systems are unable to implement the policy consistently and are unable to apply purchasing models that meet different needs, leading to a breakdown of supply chains. Further investments are needed a priori in the systems to not jeopardize the potential of free healthcare policies.

Our findings indicate that expenditure on medications remained considerable at an average of XOF2, 895 (US$4.82). This observation is consistent with prior studies, having shown that in a context of free healthcare, frequent expenses are related to medications [12]. Several factors could explain why seeking care at public facilities still results in a high expenditure on medications. For instance, previous studies have suggested that stock-outs of medicines and medical consumables at public facilities push patients to purchase drugs at private facilities, driving expenditure on drugs. According to the Ministry of Health of Burkina Faso, only 60 percent of drug orders were fulfilled by the company responsible for supplying public health facilities with essential generic drugs in 2017. This is likely to have forced patients to resort to the private sector, resulting in direct payments in spite of a free healthcare setting in the public system [16,37]. Further research is needed to understand and strengthen the supply of stocks at the Centrale d’Achat des Médicaments Essentiels Génériques (CAMEG).

Looking in-depth at the results of the two-part regression model, we noted that children have a higher probability of incurring a direct payment if they were hospitalized, live in an urban place, suffer from a severe condition, and live in the East-Central, North- Central, or West-Central region. More specifically, it is not surprising that a severe illness, since it limits daily activities and has a strong association with hospitalization, increases the likelihood of making an expenditure [11,38,39]. In line with prior studies [32,40,41], we found that OOP payments were higher among children of higher socio-economic status. This may reflect a higher ability to pay for items not directly covered by the free healthcare system, such as speciality instead of generic drugs [42,43]. Likewise, OOP payments are higher in urban settings, possibly suggesting different costs between rural and urban settings [44] and access to more opportunities for spending on additional services in urban settings not covered by the free healthcare policy. Contrary to what has been observed in other settings [45,46], we observed higher spending on children of a younger age (0 to 6 month), suggesting that there may be more gaps in the policy coverage among this age group.

It also needs to be noted that beyond direct medical costs, 44% of the children who reported out-of-pocket expenditure for care also reported spending an average of XOF728 (US$1.21) for transportation. This finding underlines the high costs associated with seeking care beyond direct medical costs. A great deal has been written highlighting distance as one of the main barriers to accessing care in Burkina Faso, as well as in other developing countries [47]. In order to fully remove financial barriers to access, policy makers need to include a lump sum to cover transport to health facilities so that distance does not impose an additional cost.

### Methodological Considerations

Beyond its policy contribution, we noted a few methodological limitations related to the nature of the data we used for our study. First, we acknowledge that the data we used were six years old by the time we submitted the article for publication. We are aware of the potential bias derived from the age of the data, as a lot could have changed in either direction, producing increases or decreases in OOP payments in Burkina Faso in the meantime. Nonetheless, we note that at the time of submission, no other large-scale population-based dataset was available in the country since the Demographic Health Survey has not been released for several years. Therefore, our estimates of OOP payments in the country remain the most relevant ones in the context of the gratuité. Second, the questionnaire was not developed to assess the determinants of spending for children aged 0 to 5 years of age [11]. As a result, information on the causes of the different sources of out-of-pocket payments is missing and could not be used to complement evidence from our analysis. Third, data were collected retrospectively, so information on out-of-pocket payments is likely to be less accurate because it is subject to recall bias [48]. Fourth, by considering children aged 0 to 5 years of age who had contact with the formal public healthcare system, we purposely truncated the sample to include only children who sought care in the public healthcare system. This is in line with our research question to determine the role of the gratuité in curbing OOP payments but may obviously lead to an underestimation of the overall magnitude of OPP for children under 5 in the country, since it captures only care encounters covered by the gratuité. Last, we recognize our inability to account for facility characteristics in our model as potential drivers of out-of-pocket expenditure due to the impossibility to link household survey data to facility data. This has limited the potential for causal inference on the role of health system characteristics in driving OOP payments under the gratuité.

## 5. Conclusions

The study revealed that a considerable proportion of children aged from 0 to 5 years, who are the target of the current free healthcare policy, continue to pay for access to healthcare. These payments are largely due to the purchase of drugs and medical consumables, probably due to stock-outs and the prescription of drugs not available in the pharmacy of public health facilities. Further research is needed to understand why these payments at point of use persist, and efforts are needed to ensure greater efficiency implementation to guarantee better financial protection.

## Figures and Tables

**Figure 1 healthcare-11-01379-f001:**
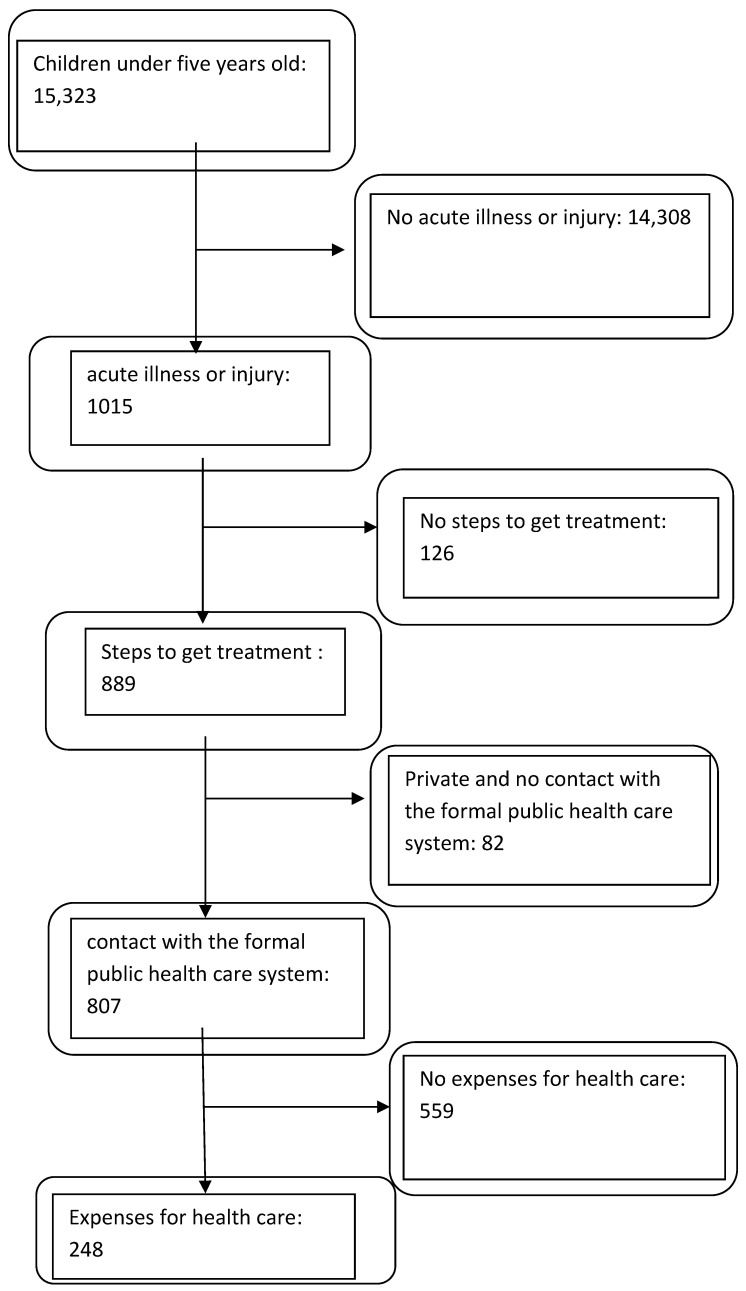
Sample identification chart.

**Table 1 healthcare-11-01379-t001:** Variables and measurements.

	Mesurements and Categorization
Variables of Interest	
Probability of out-of-pocket expenditure	0 = No; 1 = Yes
Out-of-pocket expenditure [FRS CFA]	Continuous
Explanatory variables	
Age (months)	1= (0–6); 2= (7–23); 3 =(24–59)
Sex	0 = Boy; 1 = Girl
Perceived illness severity	0 = No effect on activities; 1 = Any effect on activities
Hospitalization	0 = No; 1 = Yes
Distance from household to nearest health facility (km)	0= ≤ 5 km; 1= > 5 km
Place of residence	0 = Rural; 1 = Urban
Poverty status (SES)	0 = Other; 1 = Poor (Socio-economic status/quintile 1)
Mother’s level of education in the household	0= Uneducated; 1= Educated
District Fbr	0 = No; 1 = Yes
Region	1 = Boucle du Mouhoun; 2 = Center est; 3 = Center nord; 4 = Center ouest; 5= Nord; 6 = Sud ouest

**Table 2 healthcare-11-01379-t002:** Adjusted out-of-pocket expenditures (OOPE) ^1^ among those who used formal.

	N	%	Average (Frs CFA)	SD (Frs CFA)	Median (Frs CFA)	Min (Frs CFA)	Max (Frs CFA)
Probability of OOPE among those who used formal services (*n* = 807)							
OOP = 0	559	69	NA	NA	NA	NA	NA
OOP > 0	248	31	3407.77	2839.53	3200	100	10,000
Components of OOPE (*n* = 248)							
Consultation fees	59	24	3269.06	3459.1	1500	100	10,100
Laboratory/radiography and surgery costs (examinations)	5	2	4100	1341.64	5000	2000	5000
Drugs	237	96	2895.33	2388.14	2000	100	8000
Hospitalization costs	11	4	12,209.09	16,358.74	5050	750	50,000
Total direct health payments	248	100	3407.77	2839.53	3200	100	10,000
Transports	109	44	728.73	210.70	700	300	1000

SD: Standard Deviation. Corresponds to the costs of transport spent by those who made OOP. PS: these costs are not parts of the free health care basket.

**Table 3 healthcare-11-01379-t003:** A proportion of children aged 0–5 years who paid and who did not pay for services.

Characteristics	Children Who Paid (*n* = 248)		Children Who Did not Pay (*n* = 559)		P
	*n*	%	*n*	%	
Age groups (months)					0.012
(0–6)	58	40.28	86	59.72	
(7–23)	119	27.17	319	72.83	
(24–59)	71	31.56	154	68.44	
Perceived illness severity					0.099
No effect	92	27.54	242	72.46	
Effect	156	32.98	317	67.02	
Poverty indices					0.009
Not poor	216	32.73	444	67.27	
Poor (the poorest)	32	21.77	115	78.23	
Mother’s level of education in the household					0.708
Uneducated	195	30.42	446	69.58	
Educated	53	31.93	113	68.07	
Place of residence					0.000
Rural	206	28.22	524	71.78	
Urban	42	54.55	35	45.45	
Proximity to a health center (distance in km)					0.472
≤5 km	187	31.43	408	68.57	
>5 km	61	28.77	151	71.23	
District Fbr					0.621
No	49	29.17	119	70.83	
Yes	199	31.14	440	68.86	
Region					0.000
Boucle du Mouhoun	30	26.79	82	73.21	
Center Est	67	48.2	72	51.8	
Center Nord	44	32.35	92	67.65	
Center Ouest	38	27.74	99	72.26	
Nord	56	26.17	158	73.83	
Sud Ouest	13	18.84	56	81.16	

**Table 4 healthcare-11-01379-t004:** Two-part model for the determinants of out-of-pocket expenditure (First part: logit, second part: OLS with log transformation; *n* = 807).

	Part 1: Likelihood of Having a Direct Payment (*n* = 807)			Part 2: Determinants of the Amount of OOPE (*n* = 248)		
Explanatory Variables	OR	95% CI	*p*-Value	Coeff	95% CI	*p*-Value
Age group (months)						
(0–6 months)	1			0		
(7–23 months)	0.52	0.35–0.83	0.003 ***	−0.09	−0.39–0.19	0.51
(24–59 months)	0.62	0.39–1.01	0.052 *	−0.015	−0.33–0.30	0.92
Hospitalization						
No	1			0		
Yes	4.53	1.92–8.88	<0.001 ***	0.78	0.38–1.17	<0.001 ***
Perceived illness severity						
No effect	1			0		
Effect	1.53	1.07–2.09	0.018 **	0.24	−0.003–0.49	0.047 **
Place of residence						
Rural	1			0		
Urban	3.47	1.88–5.34	<0.001 ***	0.34	−0.032–0.66	0.031 **
Poverty Indices						
Not poor	1			0		
Poor	0.67	0.44–1.07	0.098 *	0.09	−0.25–0.45	0.61
Regions						
South-West	1			0		
Boucle du Mouhoun	1.51	0.70–3.28	0.288	−0.02	−0.63–0.58	0.945
Central-East	4.50	2.18–9.29	<0.001 ***	0.60	0.06–1.14	0.029 **
Central-North	2.43	1.16–5.08	0.018 **	0.59	0.02–1.16	0.040 **
Centre-West	1.60	0.75–3.37	0.216	0.76	0.18–1.34	0.01 **
North	1.71	0.84–3.48	0.137	0.23	−32–0.78	0.409

* Significance at 10%; ** Significance at 5%; *** Significance at 1%.

## Data Availability

Data is available on request for reasons of confidentiality and ethics. The data presented in this study are available on request from the principal investigators of the project who provided the data.

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
