# Peer review of "Do Out-of-Pocket Payments for Care for Children under 5 Persist Even in a Context of Free Healthcare in Burkina Faso? Evidence from a Cross-Sectional Population-Based Survey"

_healthcare, 2023, doi:10.3390/healthcare11101379_

Round 1

Reviewer 1 Report

In this study, the authors tried to provide an evidence on health care cost among children in relation to free health care policy in Burkina-Faso. Overall, the given evidence/table, information is narrow, and authors are unable to provide any novelty/additional knowledge related to the given topic. The methodology section is weak and did not provide comprehensive and novel information regarding research methodology/data collection and compilation. While presented results also failed to generate a convincing argument to publish the study. 

Reviewer 2 Report

The manuscript evaluated the determinants of direct payments for healthcare for children under 5 in Burkina Faso. I will discuss several issues.

Please provide rationales for selection the set of explanatory variables. For example, why the type of healthcare facility not included, as the authors pointed out, in the introduction section, that few studies have examined characteristics of patients and health facilities potentially associated with costs. In fact, almost no health facility factors were included, were these measures not available?

The presentation of the results must be improved.

-       Table 1 is not really necessary as the specification of the variables were evident in Table 2. The categories of mother’s education level were inconsistent between Tables 1 and 2.

-       Please replace all commas with decimal points – it is very confusing because commas were sometimes used to separate large numbers.  

-       Please provide n for part 1 and part 2 separately in Table 4, because these two samples were different.

-       Please report odds ratios and 95% CIs instead of the crude logit coefficient to easy the interpretation in Table 4. The labelling of perception of disease severity looks strange (“effect”), consider change.

-       Add reference groups or specify reference groups in a footnote in Table 4.

Table 2 – why not provide descriptive statistics for those incurred expenses vs those who did not? The denominator is unnecessary because all variables have the same denominator, and the N has shown in the first row. Provide unit for the variable “proximity to a health centre”.

In Table 4, the coefficient for educated mother (-1.46, SE 0.208) was not statistically significant, this is implausible given the coefficient and standard error. Please double check whether there is an error.

The interpretation of the coefficients in the Part 2 of Table 4 is insufficient. Given the log transformed dependent variable, exponentiating the coefficients and interpreting the coefficients in proportional differences of geometric means are needed.

Please provide a reference for the approach used for determining SES status (e.g., the MCA method).

In discussion section, please also put the findings of direct payment in context (e.g., in contrast with average daily wage). Also, please convert the key numbers in USD, easier for international readers to understand.

The data collection was done in 2017, almost 6 years ago. It would be helpful to discuss this limitation and provide relevant recent development of the policy in the local context.

Round 2

Reviewer 1 Report

The revised manuscript is very much improved and suitable for publication.